# Transient and Efficient Vascular Permeability Window for Adjuvant Drug Delivery Triggered by Microbeam Radiation

**DOI:** 10.3390/cancers13092103

**Published:** 2021-04-27

**Authors:** Sara Sabatasso, Cristian Fernandez-Palomo, Ruslan Hlushchuk, Jennifer Fazzari, Stefan Tschanz, Paolo Pellicioli, Michael Krisch, Jean A. Laissue, Valentin Djonov

**Affiliations:** 1Institute of Anatomy, University of Bern, 3012 Bern, Switzerland; Sara.Sabatasso@hcuge.ch (S.S.); cristian.fernandez@ana.unibe.ch (C.F.-P.); ruslan.hlushchuk@ana.unibe.ch (R.H.); jennifer.fazzari@ana.unibe.ch (J.F.); stefan.tschanz@ana.unibe.ch (S.T.); jean-albert.laissue@pathology.unibe.ch (J.A.L.); 2Biomedical Beamline ID17, European Synchrotron Radiation Facility, 38043 Grenoble, France; paolo.pellicioli@esrf.fr (P.P.); krisch@esrf.fr (M.K.)

**Keywords:** vascular permeability, drug delivery system, microbeam radiation therapy (MRT), Chicken Chorioallantoic Membrane (CAM), U-87 Glioblastoma

## Abstract

**Simple Summary:**

One of the major challenges in the pharmacological treatment of solid tumours is ensuring that therapeutic concentrations of the agent reach and penetrate the tumour tissue. This is hampered by physiological barriers imposed by the aberrant and abnormal vessel structures of the tumours and high intratumoural pressure. We show that compound penetration into tumour tissue can be greatly enhanced by irradiating the tumour with an arrangement of discrete, synchrotron generated parallel X-rays in a range of 25–50 μm in width. This irradiation geometry induces a transient increase in vessel permeability in a time-dependent manner with a maximum between 45 min and 2 h after irradiation. The latter phenomenon was fully characterized in a vascular model of the developing chick embryo and termed “permeability window”. The reported methodology could be considered as a potent and unique drug delivery system for combined tumour treatment. This will help to create new, more efficient treatment strategies against cancer and other vascular diseases.

**Abstract:**

Background: Microbeam Radiation Therapy (MRT) induces a transient vascular permeability window, which offers a novel drug-delivery system for the preferential accumulation of therapeutic compounds in tumors. MRT is a preclinical cancer treatment modality that spatially fractionates synchrotron X-rays into micrometer-wide planar microbeams which can induce transient vascular permeability, especially in the immature tumor vessels, without compromising vascular perfusion. Here, we characterized this phenomenon using Chicken Chorioallantoic Membrane (CAM) and demonstrated its therapeutic potential in human glioblastoma xenografts in mice. Methods: the developing CAM was exposed to planar-microbeams of 75 Gy peak dose with Synchrotron X-rays. Similarly, mice harboring human glioblastoma xenografts were exposed to peak microbeam doses of 150 Gy, followed by treatment with Cisplatin. Tumor progression was documented by Magnetic Resonance Imaging (MRI) and caliper measurements. Results: CAM exposed to MRT exhibited vascular permeability, beginning 15 min post-irradiation, reaching its peak from 45 min to 2 h, and ending by 4 h. We have deemed this period the “permeability window”. Morphological analysis showed partially fragmented endothelial walls as the cause of the increased transport of FITC-Dextran into the surrounding tissue and the extravasation of 100 nm microspheres (representing the upper range of nanoparticles). In the human glioblastoma xenografts, MRI measurements showed that the combined treatment dramatically reduced the tumor size by 2.75-fold and 5.25-fold, respectively, compared to MRT or Cisplatin alone. Conclusions: MRT provides a novel mechanism for drug delivery by increasing vascular transpermeability while preserving vessel integrity. This permeability window increases the therapeutic index of currently available chemotherapeutics and could be combined with other therapeutic agents such as Nanoparticles/Antibodies/etc.

## 1. Introduction

Chemotherapy is one of the most suitable treatment options for cancer therapy. However, solid tumors’ anatomical and physiological characteristics limit the exposure of all tumor cells to a sufficient concentration of such therapeutic agents [1,2]. For example, a compound must cross the vascular wall before it can affect the tumor tissue. In particular, the abnormal vasculature and the lack of a functional lymphatic network are tumor characteristics that lead to interstitial hypertension, which minimizes drug diffusion into the tumor core [3,4], ultimately diminishing the therapeutic potential of an anti-cancer compound.

Much work has focused on developing new strategies to overcome this blood-tumor barrier and improve the therapeutic potential of existing agents [5]. These strategies, which involve both pharmacological and physical approaches, include the following: Modulators of tumor blood flow reduce flow resistance through vasodilation and increase the blood pressure with vasoconstrictors, thereby also increasing the transvascular hydrostatic gradient [6]. Angiotensin II is a clinically proven agent in this group [7];Vascular normalization describes the correction of structural abnormalities by pruning immature branches, enhancing perivascular coverage, and reinstating the basal membrane [4]. This restores vascular functionality, in particular, the transportation of drugs to the tumor cells. Of all compounds used to achieve vascular normalization [8], VEGF inhibitors have been successful in clinical trials [9];Vascular permeabilization refers to the increase in capillary permeability due to the administration of inflammatory cytokines and vasomodulators, such as histamine, bradykinin, TNF-alpha, angiotensin II, botulinum neurotoxin and nitric oxide donors amongst others [5]. Some approaches use specific receptor-triggered endocytosis, i.e., employing the insulin-like growth factor 1 receptor to enable trafficking of compounds to the abluminal site [10];Overcoming the extracellular matrix (ECM) of tumors—extensive collagen networks are major obstacles for the penetration of therapeutic agents [11]. The use of collagen-degrading enzymes [12] or the downregulation of fibroblast activity [13] have shown great effectivity at improving the distribution of macromolecules;Hyperthermia is a simple, physical method that promotes drug delivery by increasing the local temperature of tissues to a range of 39–42 °C using tools such as microwaves, radiofrequency, and ultrasound. The induced capillary dilation increases perfusion and oxygenation, therefore enhancing the uptake and efficacy of chemotherapeutics [14];Ultrasound and microbubbles—the use of ultrasound in conjunction with intravenously administered microbubbles disrupts tight junction complexes and improves the delivery of chemotherapeutics in tumors [15,16]. Positive effects have been demonstrated in clinical studies of pancreatic cancer [17];Sonodynamic therapy is a novel, rapidly developing treatment based on preferential uptake of sonosensitizing compounds in tumor tissues and subsequent activation of the drug by high-intensity focused ultrasound. This strategy is minimally invasive and may be administered to deeply situated tumors [18,19].

Of all the drug-delivery systems mentioned above, more than a dozen have been approved by the Food and Drug Administration agency of the United States; however, most of them are mainly physical or only allow for topical application. This is not surprising since the permeability of the blood vessels is affected by the size and charge of the plasma components, making the delivery of macromolecules even more difficult compared to skin application [19]. As a result, there is a great need for a simple, precise, well-tolerated, and reliable drug delivery system to enhance the therapeutic potential of anti-cancer agents.

Synchrotron Microbeam Radiation Therapy (MRT) could be used as a novel drug delivery strategy that transiently enhances vessel permeability in tumors before drug administration. MRT has a unique vascular disruptive effect, where only the immature vessels are destroyed, while mature microvasculature is preserved [20,21]. MRT is based on the spatial fractionation of synchrotron-generated X-rays into arrays of micron-wide parallel, planar beamlets (25–100 µm), spaced 50–500 µm from center-to-center [22]. This generates a heterogenous dose deposition with tissue in the beam path receiving high (peak) doses of radiation (hGy) and the regions between microbeams (valley) receiving much lower doses (Gy). MRT has shown exceptional tumor control by reducing or even stopping tumor growth [23,24,25,26,27,28,29]. One potential mechanism of action involves MRT’s preferential destruction of the immature dysregulated tumor vasculature, which decreases tumor blood volume, leading to necrosis [23,30]. Remarkably, normal tissues show extremely high tolerance to MRT, as has been observed in the brains of rodents [26,31,32,33], piglets [34], duck embryos [35], cerebella of suckling rat pups [36,37], weanling piglets [34,36], and in different types of normal tissues of mice after partial or total body MRT irradiation [38,39] (recently reviewed in [22]). This normal tissue sparing effect has been attributed to the preservation of mature microvasculature. Due to the spatial fractionation of MRT, vascular damage is confined to the beam path and, unlike the tumor, the minimally irradiated endothelial cells in the valley region can repair neighboring regions damaged by the microbeam [33]. The unique vascular disruptive effects of MRT have been demonstrated in the Chick Chorioallantoic Membrane (CAM) where the vascular properties resemble those of tumors [20]. The CAMs were exposed to peak doses of 200–300 Gy, which preferentially destroyed immature vessels (with the first subcellular changes occurring 15 min after exposure) while vascular integrity was maintained in the valley regions. This promoted the resolution of damaged regions and subsequent clearance of edema one hour post-irradiation with sustained capillary perfusion. Perfusion studies 6 h following MRT with FITC-dextran showed zones of intact, perfused capillaries in the valley (low-dose) region and vascular disruption and loss of perfusion limited to the microbeam path (high-dose) [20]. The reversible damage to the vasculature is attributable to the spatial fractionation of the incident beam, as homogenous dose delivery resulted in unresolved damage at doses hundreds of Gy below those delivered by MRT. The induction and subsequent resolution of edema suggest that MRT induces a transient increase in vascular permeability following MRT that could be exploited for therapeutic gain.

To further explore and characterize this observed period of transpermeability, we evaluated the effects of delivering MRT at a peak dose below the threshold in which vessel destruction was observed with the goal of preserving perfusion to the tumor and tumor-like vessels of (1) the vascularized CAM and (2) subcutaneous xenografts of human glioblastoma.

## 2. Materials and Methods

### 2.1. Animal Models

Two animal models were used for these experiments: the chick CAM and human U-87 Malignant Glioma xenotransplanted in BALB/c nude mice.

*CAM:* Fertilized chick eggs from a commercial hatchery were transferred to Petri dishes on the third day after fertilization following the shell-free culture method [40]. Embryos were maintained at 37 °C under a humidified atmosphere until day 12 of the embryonic development.

The CAM is the extraembryonic network of rapidly developing vasculature supporting respiration of the developing chicken embryo. Due to the ease of visualization and rapid development, the CAM model has been used extensively in the field of angiogenesis research with each stage of embryonic development corresponding with various stages of vascular maturation. This rapid vessel development also supports tumor grafts for the study of tumor dynamics, without the need for costly rodent models and eliminating ethical concerns [41]. The CAM has therefore become an indispensable model for the study of vessel development, and dynamics in particular, when testing anti-angiogenic therapies [42]. The versatility of the CAM as an experimental model has been extensively reviewed [43,44,45]. One of the major benefits of this model is that it can be maintained ex-ovo, permitting the real-time observation of vascular changes in response to various targeted treatments including radiation therapy (review by Mapanao et al., 2021 [46]). This made it an ideal system for visualizing MRT-induced changes in vascular permeability.

Human glioblastoma xenografts: U-87 Malignant Glioma cells (ECACC, Salisbury, UK) were cultured in D-MEM supplemented with 10% fetal calf serum and 1% antibiotics/antimycotics. Tumor cells (2 × 10^6^ in 100 µL PBS per mouse) were implanted subcutaneously (sc) in the right flank of 61 male BALB/c nude mice, weighing about 20–22 g (Charles River Laboratories, Paris, France). There were four experimental groups: Control group (CO, *n* = 13), Cisplatin-treated group (CIS, *n* = 17), MRT group (MRT, *n* = 17), and double-treated group (MRT + Cisplatin, *n* = 14). A dose of 10 mg/kg BW of Cisplatin (Cisplatin-Teva^®^, Teva Pharma AG, Basel, Switzerland) was administered via the tail-vein 40 min after MRT.

### 2.2. Synchrotron Microbeam Irradiation

The irradiations of both animal models were performed at the ID17 Biomedical Beamline of the European Synchrotron Radiation Facility (ESRF) in Grenoble, France.

*CAM:* CAMs at day 12 of development were irradiated with a 1 × 1 cm array of 51 microbeams of 25 µm in width on average, spaced by 200 µm from their centers. To achieve this array configuration, we used a multi-slit “Archer” collimator with alternating Au and Al foils and fixed geometry [47,48]. A wiggler gap of 40 mm delivered an X-ray spectrum configuration of 93.4 keV mean energy, and 74.9 keV peak energy. Peak-entry doses were estimated at 75 Gy according to our Monte Carlo computation. A radiochromic film (GafChromic^®^ radiochromic film type HD-810, ISP Corporation, Wayne, NJ, USA) was laid over the surface of the CAM prior to the irradiation, to visualize the microbeam paths and distinguish between the irradiated and non-irradiated parts.

*Glioblastoma xenografts*: Mice were anesthetized with an intraperitoneal injection of xylazine/ketamine (0.1/1% in saline solution buffer, 10 µL/g body weight), then placed on their left flank on a horizontal surface. The tumors were irradiated seventeen days after tumor cell inoculation when their volumes averaged 200 mm^3^ (calculated by the formula: V(mm^3^) = 4/3 × π × a × b × c, where a, b and c are the length, the width, and the height of the tumor, measured by digital caliper). The tumors were irradiated unidirectionally with a skin-entrance dose of 150 Gy, using 50 µm wide microbeams and 200 µm on-center distance. This configuration was achieved by the ESRF-made multi-slit collimator made of tungsten carbide [49]. A wiggler gap of 24.8 mm delivered an X-ray spectrum of 104.2 keV mean energy, and 87.7 keV peak energy. The average dose rate was 12,000 Gy/s.

### 2.3. Vascular Permeability Assay with FITC-Dextran

Microscopic observations were made up to 48 h after MRT with video documentation. For the assay, we used FITC-dextran of MW 2 × 10^6^ Daltons, (Fluorescein isothiocyanate from Sigma, Taufkirchen, Germany). FITC-dextran has a Stroke Ratio of 270 Angstrom [50], which converts to approximately 27 nm.

We also employed red fluorescent polystyrene microspheres (FluoSpheres^TM^ Carboxylate-Modified Microspheres, Cat #: F8810, ThermoFisher, Waltham, MA, USA). The microspheres have a diameter of 100 nm with a coefficient variation in size of 5% [51]. Their surfaces are pre-coated with a high density of carboxylic acids, which endows the microspheres with a highly charged and relatively hydrophilic surface layer. The surface charges range between 0.1 and 2.0 mEq/g, which makes them stable to a relatively high concentration of electrolytes (max. 1 M univalent salt) and prevents agglomerates [52]. We selected this microsphere because the highly charged surface reduces their attraction to cells, which makes them ideal for studying vascular permeability.

*CAM:* CAMs were either left untreated or intravenously injected with FITC-dextran and red fluorescent microspheres. Moreover, we applied 1.0 µg of recombinant VEGF-A165 protein (Peprotech, London, UK) on the surface of 7 CAMs, and compared them against VEGF-untreated CAMs. A semi-quantitative analysis of the extravasated FITC-dextran was performed in vivo in the CAM. The blood flow and FITC-dextran extravasation were monitored every 15 min. The intensity of perivascular FITC-dextran was evaluated according to the score presented in the caption of Figure 1. The video sequences documented the presence and site of the microbeam stripes, the optically empty zones, the damaged medium- and large-sized vessels, the damage to the capillary network, and extravasation of the fluorescent probes.

*Glioblastoma Xenograft:* Forty-five minutes after irradiation, we performed a vascular permeability assay in a selected group of mice; MRT + Cisplatin (*n* = 4), MRT alone (*n* = 7), Cisplatin alone (*n* = 4), and control (*n* = 4). A solution of 3% FITC-dextran in sterile saline was injected (0.3 mL) into the tail vein of mice. In all cases, tumors were harvested approximately 30 min after injection of the fluorescent compound and fixed in 2% paraformaldehyde.

### 2.4. Semi-Thin Serial Sectioning and Transmission Electron Microscopy (TEM) of CAM

The sites of interest from CAM samples were harvested and fixed in 2.5% (*v*/*v*) glutaraldehyde solution buffered with 0.03 M potassium phosphate (pH 7.4, 370 mOsm), post-fixed in 1% OsO_4_ (buffered with 0.1 M sodium cacodylate (pH 7.4, 340 mOsm)), dehydrated in ethanol, and embedded in epoxy resin. Thousands of 0.8 µm-thick serial sections, perpendicularly to the direction of the beam propagation, were prepared with glass knives and stained with toluidine blue. The serial sections were then viewed, and images captured at different magnifications using a light microscope (Leica, Leitz DM, Morrisville, NC, USA), equipped with a Leica DFC480 camera. For transmission electron microscopy, 80 to 90 nm-thick sections were prepared and mounted on copper grids coated with Formvar (polyvinyl formal; Fluka, Buchs, Switzerland). They were stained with lead citrate and uranyl acetate and viewed in a Philips EM-400 electron microscope [53].

### 2.5. Immunostaining and Analysis of Glioblastoma Xenograft

Sections of tumor blood vessels were dewaxed, rehydrated, and subjected to heat-induced epitope retrieval (Dako S1699). Endogenous proteins were blocked with 5% BSA in PBS for 30 min at room temperature, followed by the primary antibody (rabbit anti-mouse CD31, Abcam, Eugene, OR, USA), diluted 1:20 in 1% BSA/PBS and incubated for 48 h at 4 °C in a humidified chamber. After washing with PBS-Tween 20 for 30 min, sections were incubated with goat anti-rabbit Alexa Fluor 594 (Invitrogen, Carlsbad, CA, USA) secondary antibody at a dilution of 1:200 in 10% FCS/PBS for 30 min at room temperature. Quantification of the extravasated probe was performed, based on pictures taken with a confocal microscope (LSM Zeiss Meta, Caochen, Germany). For each experimental group, three to seven pictures per tumor were taken with three-vessel areas (VA) measured per picture. The overall fluorescence (OF) and the intravascular fluorescence (IF) of FITC-dextran were quantified using the ImageJ software. The extravascular fluorescence (EF) was defined as OF-IF. The vascular permeability index was calculated as EF/VA.

### 2.6. Magnetic Resonance Imaging (MRI) of Glioma Xenografts

A selected group of mice underwent MRI on days 0, 5, 13, 20, and 27 post-treatment to monitor tumor growth. Each group had 5 mice, including the control group. MRI was performed with a 4.7 Tesla Scan (Avance III console, Bruker, Ettlingen, Germany) at the “Institut des Neurosciences” in Grenoble, France. The animals were subjected to an anatomical T1/T2 scan, and to a permeability MRI after an intravenous injection of gadolinium-labelled albumin (Gd-Albumin, BioPAL) via the caudal vein.

## 3. Results

### 3.1. Microbeams Induced a Transient Vascular “Permeability Window” in CAM without Impairing Tissue Perfusion

After irradiating the CAM with microbeam entrance peak-doses of 75 Gy, there was an increase in transpermeability without vascular destruction and preservation of vascular perfusion. The vascular permeability assay revealed that only FITC-dextran (~27 nm) extravasated into the surrounding tissue, while the larger microspheres (100 nm) remained stuck along the microbeam paths (Figure 1a,b). Successive semi-quantitative evaluation (every 15 min) showed that the extravasation of FITC-dextran was transient, detected from 15 min until it ended at 4 h after irradiation (Figure 1f, Appendix A).

In addition, we administered VEGF to the top of the CAM prior to irradiation to induce neovascularization. The goal was to simulate the tumor microenvironment, which normally has high amounts of VEGF. Then, we compared the vascular effects caused by MRT in the VEGF-treated (Figure 1c, left side) and non-treated areas (Figure 1c, right side). We found that the vascular transpermeability occurred earlier (10 min after irradiation) in the VEGF-induced neovasculature (higher magnification in Figure 1d) than in the VEGF-untreated vasculature (higher magnification in Figure 1e). This suggests that immature vessels are more sensitive to the microbeams and show an earlier onset of vascular transpermeability.

### 3.2. Time Course of the Structural Changes in the CAM during the Vascular “Permeability Window”

Fifteen minutes after microbeam radiation of 75 Gy, the CAM thickness increased transiently to approximately three times its regular size. This was assessed by comparing it against the recovered CAM 4 h post-irradiation (Figure 2a,d). The acute increase in size is likely attributed to the development of edema underneath the capillary plexus (Figure 2b). The ultrastructural analysis revealed a discontinuous luminal surface of the microvessels, with rarefication of the endothelial cytoplasm resulting in fissures and gaps. The increased permeability was evidenced by the presence of FITC-dextran dots in the endothelial cell wall of the microvessels, as well as in the extravascular space (Figure 2c(c^1^,c^2^)). Conversely, 4 h after irradiation, the endothelial cells showed restored integrity, which was accompanied by only single holes and solitary FITC-dextran depositions in the endothelium (Figure 2f(f^1^)). These observations suggest that microbeams of 75 Gy increased the vascular transpermeability without long-lasting damage to CAM vasculature.

### 3.3. Microbeams also Induced Vascular Permeability in Human U-87 Glioblastoma Xenografts

To determine whether microbeams promote vascular permeability in a human glioblastoma xenograft mouse model, we compared one group treated with 150 Gy (peak-entry dose) of microbeams with an unirradiated tumor control group (Figure 3). We observed clear extravasation of FITC-dextran in the irradiated tumors 45 min post-irradiation (Figure 3d). Conversely, the fluorescent compound remained intravascular in the control group (Figure 3c). The permeability index revealed a two-fold increase in transpermeability following MRT relative to the unirradiated control (Figure 3e). At the ultrastructural level, no extravasation of the fluorescent probe was observed in control tumors; FITC-dextran dots remained in the lumen (Figure 3f,h). However, in microbeam-treated tumors, FITC-dextran was observed in the extravascular space together with partially disintegrated endothelial cells (Figure 3g,i). These results confirm that MRT can also induce vascular permeability in this mammalian tumor model, and thus, vascular permeability is not restricted to the CAM (avian).

### 3.4. Using the MRT-Induced Vascular Permeability to Enhance the Delivery of Cisplatin

To exploit the MRT-induced “permeability window”, the adjuvant Cisplatin was administered in conjunction with MRT in mice bearing glioblastoma xenografts. Cisplatin is known to have efficacy against glioblastoma in vitro but a poor clinical response as a single agent and in combination with radiotherapy [54]. This is primarily due to poor penetration across the blood–brain barrier [55] and dose-limiting cytotoxicity [56]. Microbeam radiation therapy was delivered to the tumors 17 days after cell inoculation, and Cisplatin was administered 40 min after irradiation. Tumors in the control group began to grow exponentially 2 days after treatment (Figure 4). Differences between the treatment groups started on day 13, with the fastest-growing tumors belonging to the Cisplatin group, followed by those treated with MRT alone. In contrast, tumors treated with the combination of MRT + Cisplatin remained unchanged until approximately 22 days after treatment, when their growth rate began to abate slowly. In a second experimental trial, tumor growth measurements performed with Magnetic Resonance Imaging (MRI) on days 0, 5, 13, 20 and 27 after treatment yielded tumor volumes comparable to those measured with the digital caliper; tumor volumes decreased in the same order: Control > Cisplatin alone > MRT alone > MRT + Cisplatin (Figure 5a). Accordingly, images of tumor progression (Figure 5b) show the best treatment results on animals subjected to MRT + Cisplatin in comparison with the other experimental groups; with a 2.75-fold decrease in comparison with MRT alone, and a 5.25-fold decrease compared to Cisplatin alone. These results show that the administration of adjuvant Cisplatin can take advantage of the MRT-induced vascular permeability.

## 4. Discussion

One of the main challenges in cancer treatment is the delivery of sufficient quantities of chemotherapeutic agents and nanocarriers to tumors. For instance, many compounds do not extravasate into normal tissue but passively cross leaky tumor capillaries in a process referred to as Enhanced Permeability and Retention (EPR) [57]. This EPR has been shown to allow the passage of molecules from 40 to 70 kDa [58]. However, a retrospective study reported that, unfortunately, only 0.7% (median) of the administered agents were delivered to a solid tumor [59]. Furthermore, the chemotherapeutic compounds are often small molecules with a short half-life, and multiple applications or higher doses are needed to increase their therapeutic impact [60]. This can result in severe negative side effects and possible drug resistance [61]. Many of the approaches mentioned in the introduction have different limitations, and only a few are actually applied clinically. For example, modulators of blood flow and vascular normalization have a short half-life, making it difficult to estimate in advance the dose and timing of the drug administration [62]. Furthermore, systemic administration of inflammatory cytokines and vasomodulators for vascular permeabilization are associated with high, whole-body toxicity. For this reason, they are used in the clinic only for isolated limb perfusion, e.g., for the treatment of sarcomas and melanomas [63]. Therefore, to improve the therapeutic index of the wide range of potent anti-cancer agents, we need a precise, accurate, and well-tolerated drug delivery system that can increase their delivery into tumors while minimizing damage to normal tissue.

The promising new strategy for the enhancement of vessel permeability presented here may overcome the obstacles of the blood–tumor barrier (BTB). Low-dose MRT is a very simple and highly effective physical solution that does not rely on the use of any carriers. It induces a transient, vascular permeability window in CAM, beginning 15 min after MRT (75 Gy) and ending at 4 h. The extravasation of FITC-dextran (MW 2 × 10^6^ Dalton, size of 27 nm) was visible as green “clouds” diffusing between high-dose microbeam regions, which indicates a penetration depth of a few hundreds of micrometers. At the same time, larger microspheres of 100 nm were constrained to the beam path (Figure 1a,b,f, and Appendix A), indicating that the transpermeability of particles following MRT is size-dependent.

Previously, it has been shown that diffusion of the small fluorescent probe, sulforhodamine B (0.58 kDa), across the blood–tumor barrier in 9 L gliosarcoma was only induced when peak doses were delivered at 1000 Gy and persisted up to 30 days post-irradiation [31]; meanwhile, diffusion of a larger 70 kDa FITC-dextran was not observed at any dose or time-point. However, the permeability observed by these authors was at much later time-points, with measurements beginning only 12 h following MRT (instead of at 15 min as we present in this manuscript), and the permeability was likely a consequence of endothelial destruction caused by the 1000 Gy peak dose [64]. Furthermore, an MRT scheme of 2 cross-fired arrays, each delivering a peak dose of 400 Gy (total dose of 800 Gy to the tumor), was effective in the destruction of tumor tissue but also induced edema in normal brain tissue, causing the majority of animal deaths in that study [30]. Although both studies acknowledge the use of MRT to increase permeability, the late time points of observation missed the effective “permeability window” that we report in this manuscript. Moreover, these studies used high peak doses that triggered the consequent vascular damage, while we show evidence that the delivery of lower MRT peak doses (<150 Gy) can enhance drug delivery to the tumor without inducing endothelial damage nor subsequent pathology. This is evidenced in the ultrastructure observations of endothelial cells from irradiated CAMs and glioma xenografts (Figure 2 and Figure 3). Four hours after MRT, the vascular integrity was almost completely restored, and the permeability window closed (Figure 1).

Many tumors secrete high concentrations of VEGF, which is known to increase the hyperpermeability in already existing microvessels and at the same time induce neoangiogenesis in rapidly growing tumors [65,66]. Specifically, VEGF promotes vascular permeability by disrupting endothelial cell contacts. Exogenous VEGF administration to the CAM induced the rapid permeability of CAM vessels following MRT in comparison to MRT administered alone, decreasing the onset time of the permeability window from 15 min to 10 min (Figure 1c–e). This additive effect is a promising attribute of tumors producing a high amount of VEGF when they are treated by microbeams. MRT itself at higher peak doses has been shown to induce VEGF expression in normal and tumor tissue in the brain over time, contributing to brain edema [30]. However, lower doses of MRT (150 Gy), as used here, have been shown to induce a vascular normalization effect, with increased pericyte coverage and resolution of hypoxia within 2 weeks of treatment [28]. In previous studies, we reported that MRT at higher doses, in the range of 300–400 Gy, had a preferential vascular destructive effect in both the chick CAM vasculature [20] and the murine model of melanoma [29]. It seems that after the destruction of capillaries, the width of the microbeam is essential for the prediction of the grade of restoration of their integrity. In another study, we partially amputated the ventral half of the caudal fin of zebrafish to induce regeneration and the development of new, immature vasculature that mimics that seen in a tumor. After regeneration, we irradiated the regenerating ventral (immature) and undamaged, dorsal (mature) compartments with 25, 50, 100, 200, and 800 µm wide beamlets [21]. The restoration of vascular defects was observed when the beamlets were up to 100 µm wide, but not when they were 200 or 800 µm wide. It has been shown that one of the major mechanisms of the action of MRT is mediated by the induction of vascular toxicity on immature, or tumoral microcirculation (reviewed in [22]). The geometry of MRT, together with the applied doses, appears to be essential for the differential endothelial disintegration, which may be species-specific.

In the second part of this study, we exploited this “vascular permeability window” to enhance the delivery of co-adjuvant Cisplatin into U87 human glioblastoma xenografts in mice. First, it was confirmed that MRT could also induce a vascular permeability window in the glioblastoma xenograft model, as shown by the extravasated FITC-dextran 45 min after 150 Gy of MRT (Figure 3). Secondly, we showed the treatment efficacy of combining 150 Gy MRT + Cisplatin in two separate trials, where tumor measurements were performed with a digital caliper or more accurately with MRI, in each respective trial. The combined treatment of 150 Gy MRT + Cisplatin achieved the best tumor control among all treatment groups (Figure 4 and Figure 5).

However, despite showing vascular permeability in glioblastoma xenografts with FITIC dextran (Figure 3), the present survival study had the limitation of not allowing for the measurement of the accumulated cisplatin in the tumor. Future mechanistic studies should include this variable to confirm the vascular permeability hypothesis.

## 5. Conclusions

In conclusion, we have confirmed that low doses of 75 and 150 Gy (relative to the MRT field) increase the vessel permeability in the chick CAM and a glioma xenograft model, respectively. Besides the effect of MRT on tumor growth, the results suggest that a preceding exposure to microbeams may render tumors more accessible to drug delivery. The MRT-induced vascular permeability observed in glioma xenografts could be exploited for the treatment of other intracranial tumors for which the transport of chemotherapeutic agents through the blood-brain barrier is difficult and limits treatment success [67]. Finally, the transient vascular permeability induced by MRT could also be applied to the delivery of drugs and/or agents other than chemotherapeutics, such as nanoparticles, antibodies, or vectors for the treatment of tumors or other pathologies.

## Figures and Tables

**Figure 1 cancers-13-02103-f001:**
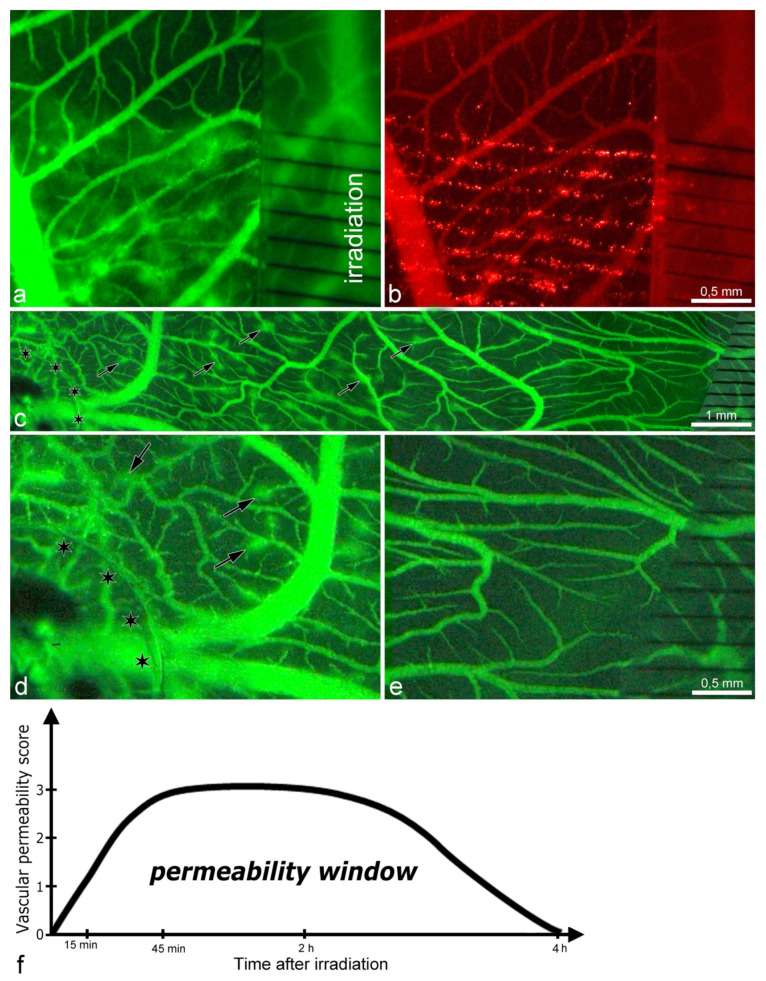
Images from intravital microscopy showing increased vascular permeability in CAM after exposure to MRT peak doses of 75 Gy; (**a**,**b**): normal CAM vasculature; (**c**–**e**): vasculature after VEGF treatment. Note: (**a**) forty-five minutes after exposure to MRT, the vascular permeability is increased, as demonstrated by the extravasation of FITC-dextran (green-fluorescent halos around the blood vessels). Conversely, in (**b**), the microspheres did not diffuse into the surrounding tissue but remained affixed as red-fluorescent dots along the microbeams path. Left side: (**c**) at the site of VEGF application (asterisks indicate the edge of the Thermanox^®^ coverslip), the vascular permeability increased as early as 10 min after MRT, as shown by the halos of extravasated FITC-dextran (some marked by arrows). Right side: in the non-treated zone, no such signs of increased vascular permeability were observed simultaneously. Images (**d**,**e**): parts of (**c**) at higher magnification. Image (**f**): schematic representation of the vascular permeability window after 75 Gy of MRT. The score: (0) = no FITC extravasation; (1) = small non-confluent FITC “halo” surrounding the capillaries; (2) = FITC “halos” start merging but they are not completely confluent; (3) = the “halos” are completely confluent. In (**a**–**e**), black stripes on the radiochromic film indicate the path of the microbeams.

**Figure 2 cancers-13-02103-f002:**
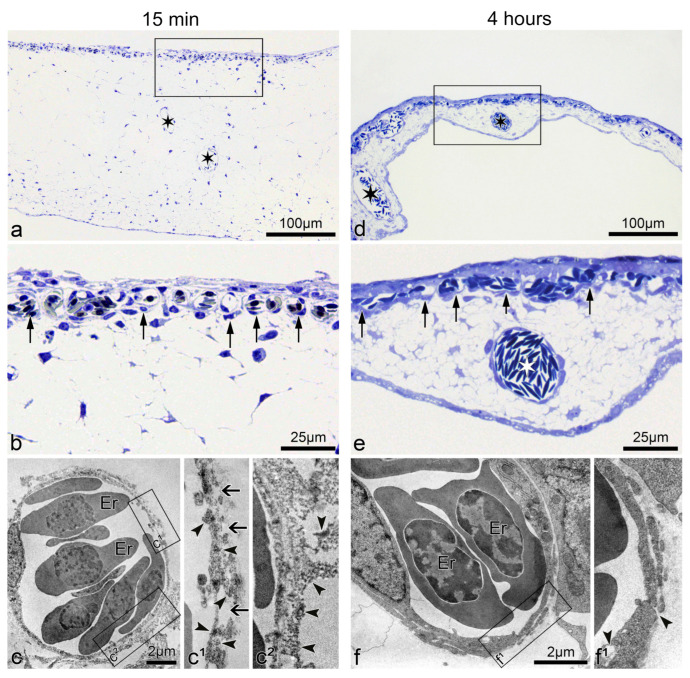
Morphological alteration of CAM vasculature 15 min and 4 h post-75 Gy of MRT. Images (**a**,**b**): semithin section of CAM fifteen minutes after MRT exposure: Irradiated CAM is enlarged (edematous). The capillary vessels (arrows) appear almost normal at light microscopy. (**c**(**c^1^**,**c^2^**)): Ultrastructure of CAM samples shown in (**a**,**b**) reveal a discontinuous endothelium with gaps and fissures (arrows). Those are most likely responsible for the increased permeability, as demonstrated by the presence of FITC-dextran dots (arrowheads) in the endothelial cell wall (**c^1^**) as well as in the abluminal space (**c^2^**). Images (**d**,**e**): four hours after 75 Gy of MRT, the CAM thickness decreased, thus almost reverting to the normal morphology. The capillary plexus (arrows) and supplying vessels (white asterisk) appear perfused and intact in semithin sections. Images (**f**(**f^1^**)): four hours after microbeam exposure, the capillaries regained their normal ultrastructure, as evidenced by the nearly normal endothelial cells. Only occasional vacuoles and fissures were present (arrowheads). Images (**b**,**c^1^**,**c^2^**,**e**,**f^1^**) are higher magnifications of the rectangles in (**a**,**c**,**d**,**f**), respectively. Er = erythrocyte.

**Figure 3 cancers-13-02103-f003:**
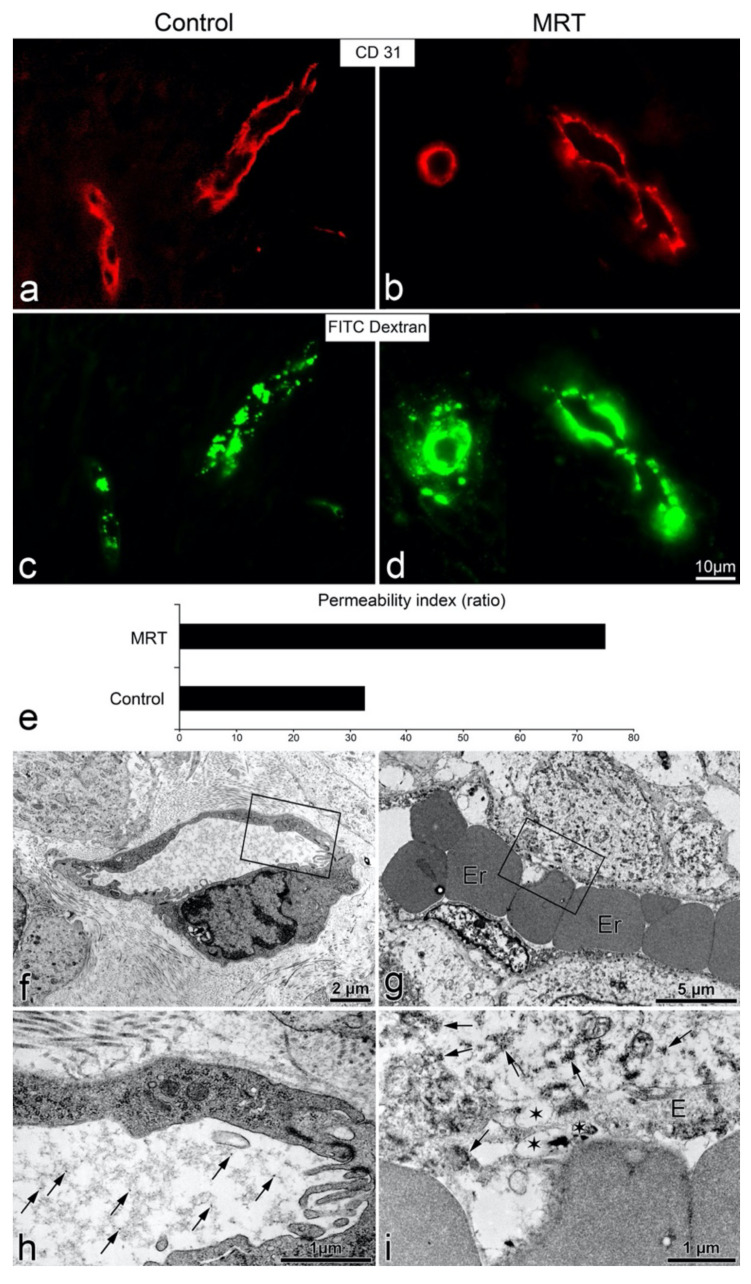
MRT-induced vascular permeability in mice harbouring the human U-87 glioblastoma xenograft. Fluorescence microscopy for CD31 and FITC-dextran in a control tumor (**a**,**c**); tumor post-MRT (**b**,**d**). There was no extravasation of the green FITC-dextran in the control tumor (**c**), while in the MRT-treated tumor (**d**), 45 min after 150 Gy, a bright halo of green fluorescence was visible. Image (**e**): graph showing the quantification of the vascular permeability in controls and MRT-treated tumors as the ratio of extravasated FITC-dextran fluorescent area/vessel area. The ultrastructure of tumor vessels was normal in controls (**f**,**h**), with no extravasation of FITC-dextran (intraluminal dextran as dark dots indicated by arrows). Conversely, in treated tumors (**g**,**i**), an extravasated fluorescent probe material was observed as dark dots (arrows) in the extravascular space; the disrupted endothelium contained multiple vacuoles of different sizes, indicated by asterisks. Er = erythrocyte. Images (**h**,**i**) are higher magnifications of the rectangles in (**f**,**g**), respectively.

**Figure 4 cancers-13-02103-f004:**
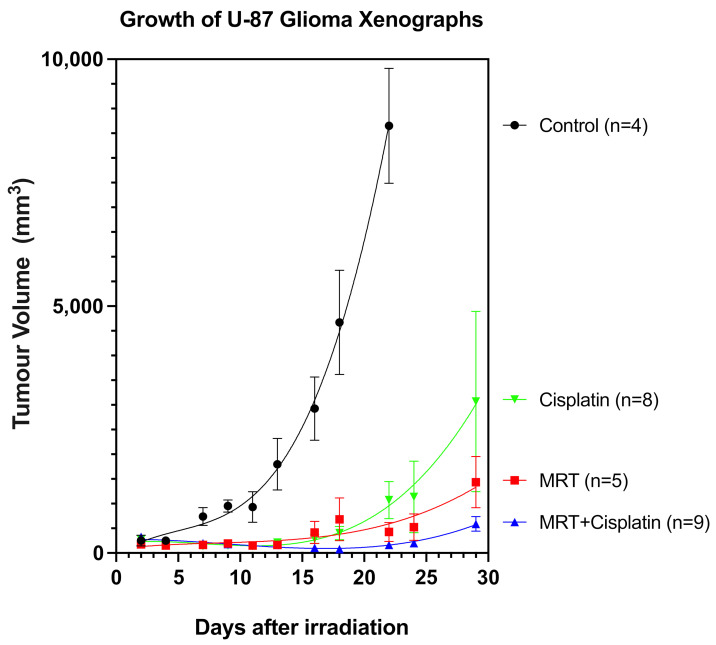
Growth of U-87 glioblastoma xenograft. Groups are unirradiated Controls (*n* = 4), Cisplatin (*n* = 8), MRT (*n* = 5), and Double Treatment (MRT + Cis) (*n* = 9). The tumors were measured with a digital caliper every second day.

**Figure 5 cancers-13-02103-f005:**
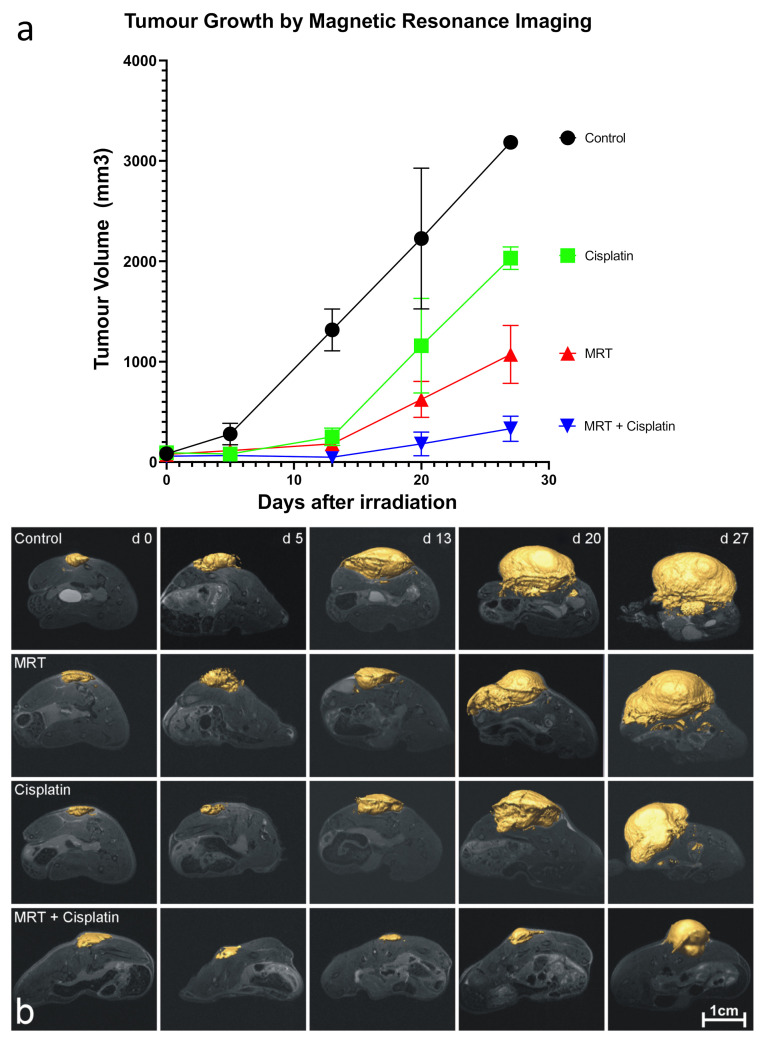
U-87 glioblastoma growth followed by MRI. Image (**a**) shows the tumor volume growth measured by MRI. Image (**b**): MRI images of the tumor progression for each animal group. Control (*n* = 3), Cisplatin (*n* = 4), 150 Gy MRT (*n* = 7), and Double Treatment (150 Gy MRT + Cis) (*n* = 5).

## Data Availability

Data are available from the corresponding author upon request.

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
