# Peer review of "Transient and Efficient Vascular Permeability Window for Adjuvant Drug Delivery Triggered by Microbeam Radiation"

_cancers, 2021, doi:10.3390/cancers13092103_

Round 1
Reviewer 1 Report
In their manuscript "Transient and efficient vascular permeability window for adjuvant drug delivery triggered by microbeam radiation," the authors highlight the potential of microbeam radiation therapy to improve drug delivery to tumors. Below are some comments on the manuscript:
- the figure legends need to be revised because figure number are missing.
- Could you please provide information regarding the x-ray energy distribution or maximum energy in the materials and methods section?
- In Figure 5 (I am not sure of the number because the figure number is missing), why are you comparing 400 Gy MRT and 150 Gy MRT + cisplatin? Why this change in delivered dose between your 2 conditions?
- Your main conclusion is that low doses of microbeam radiation therapy increase vessel permeability and thus improve the tumor loading of chemotherapy drugs. However, you do not provide information regarding cisplatin accumulation in the tumor (via a measurement of platinum concentration in tumors, for example). Moreover, you consider the doses of X-rays you use (about 150 Gy) as "low doses". However, this is far from the usual clinical dose of 2 Gy that is delivered daily to patients in radiotherapy. Although your data opens up new avenues in the design of chemo-radiotherapy treatments, it will be interesting to study your proof of concept in more relevant dose ranges (< 10 Gy).
Author Response
REVIEWER 1
In their manuscript "Transient and efficient vascular permeability window for adjuvant drug delivery triggered by microbeam radiation," the authors highlight the potential of microbeam radiation therapy to improve drug delivery to tumors. Below are some comments on the manuscript:
- the figure legends need to be revised because figure number are missing.
- Thank you. This has been fixed.
- Could you please provide information regarding the x-ray energy distribution or maximum energy in the materials and methods section?
- Thank you. We have completed the manuscript with the requested information about the spectrum for both animal models.
- In Figure 5 (I am not sure of the number because the figure number is missing), why are you comparing 400 Gy MRT and 150 Gy MRT + cisplatin? Why this change in delivered dose between your 2 conditions?
- We apologized for this confusion. The 400 Gy was a typo. All doses to the glioblastoma were 150 Gy.
- Your main conclusion is that low doses of microbeam radiation therapy increase vessel permeability and thus improve the tumor loading of chemotherapy drugs. However, you do not provide information regarding cisplatin accumulation in the tumor (via a measurement of platinum concentration in tumors, for example). Moreover, you consider the doses of X-rays you use (about 150 Gy) as "low doses". However, this is far from the usual clinical dose of 2 Gy that is delivered daily to patients in radiotherapy. Although your data opens up new avenues in the design of chemo-radiotherapy treatments, it will be interesting to study your proof of concept in more relevant dose ranges (< 10 Gy).
- We thank the reviewer for this important remark. We completely agree that adding measurements of cisplatin in-situ would enhance the vascular permeability hypothesis. For this, a mechanistic study would be necessary. We have added this as a limitation in the discussion.
“However, despite of showing vascular permeability in glioblastoma xenografts with FITIC dextran (Fig 3), the present survival study had the limitation of not allowing for the measurement of the accumulated cisplatin in the tumour. Future mechanistic studies should include this variable to confirm the vascular permeability hypothesis.”
- Regarding the “low dose” point we agree with the reviewer that from a Clinical perspective, 150 Gy are not low doses. However, we would like to respectfully point out that in the clinic 2 Gy are delivered to a volume of tissue of a few centimeters. In contrast, MRT is spatial fractionated radiotherapy where each microbeam is 50 µm wide (hundreds and even thousands of times smaller than a homogenous field of conventional radiotherapy). Therefore, in MRT, we must deliver doses that might seem “high” from a clinical perspective but are complete reasonable and necessary in Spatially Fractionated Radiation Therapy.
Historically, research on MRT has delivered doses of up to 1900 Gy. Members of our group recently performed the first scoping review of all animal models employed in MRT and the most used doses are around 400-650 Gy. We invite the reviewed to take a look a Table 1 in this review, which will show that 150 Gy are in the very “low range” of peak doses employed in the MRT field.
Fernandez-Palomo, Cristian, Jennifer Fazzari, Verdiana Trappetti, Lloyd Smyth, Heidrun Janka, Jean Laissue, and Valentin Djonov. “Animal Models in Microbeam Radiation Therapy: A Scoping Review.” Cancers 12, no. 3 (February 25, 2020): 527. https://doi.org/10.3390/cancers12030527.
Bibliography
- Potez, M.; Fernandez-Palomo, C.; Bouchet, A.; Trappetti, V.; Donzelli, M.; Krisch, M.; Laissue, J.; Volarevic, V.; Djonov, V. Synchrotron Microbeam Radiation Therapy as a New Approach for the Treatment of Radioresistant Melanoma: Potential Underlying Mechanisms. Int. J. Radiat. Oncol. 2019, 105, 1126–1136, doi:10.1016/j.ijrobp.2019.08.027.
Reviewer 2 Report
The Authors of this ms have demonstrated that CAM exposed to MRT exhibited vascular permeability, which 20 started 15 min post-irradiation, reached its peak from 45 min to 2 h, and ended at 4 h )"permeability window"). Morphological analysis showed partially fragmented endothelial walls as the cause of the increased transport of FITC-Dextran into the surrounding tissue and the extravasation of 100 nm microspheres. In the human glioblastoma xenografts, MRI measurements showed that the combined treatment reduced the tumour size by 2.75-fold and 5.25-fold, respectively, to MRT or Cisplatin alone.The Authors concluded that MRT provides a novel mechanism for drug delivery by increasing vascular spermeability while preserving vessel integrity. The permeability window increases therapeutic index of chemotherapeutics and could be combined with other therapeutic agents.
REMARKS. This is well done, original and interesting work. I suggest to the Authors to further discuss the use of the CAM assay in the study of tumor angiogenesis with the advantages and disadvantages of this assay.
Author Response
REVIEWER 2:
The Authors of this ms have demonstrated that CAM exposed to MRT exhibited vascular permeability, which 20 started 15 min post-irradiation, reached its peak from 45 min to 2 h, and ended at 4 h )"permeability window"). Morphological analysis showed partially fragmented endothelial walls as the cause of the increased transport of FITC-Dextran into the surrounding tissue and the extravasation of 100 nm microspheres. In the human glioblastoma xenografts, MRI measurements showed that the combined treatment reduced the tumour size by 2.75-fold and 5.25-fold, respectively, to MRT or Cisplatin alone.The Authors concluded that MRT provides a novel mechanism for drug delivery by increasing vascular spermeability while preserving vessel integrity. The permeability window increases therapeutic index of chemotherapeutics and could be combined with other therapeutic agents.
REMARKS. This is well done, original and interesting work. I suggest to the Authors to further discuss the use of the CAM assay in the study of tumor angiogenesis with the advantages and disadvantages of this assay.
- Thank you very much for the kind comments. We have expanded our discussion regarding the CAM as requested. Please see below:
“The CAM is the extraembryonic network of rapidly developing vasculature supporting respiration of the developing chicken embryo. Due to the ease of visualization and rapid development, the CAM model has been used extensively in the field of angiogenesis research with each stage of embryonic development corresponding with various stages of vascular maturation. This rapid vessel development also supports tumour grafts for the study of tumour dynamics without the need for costly rodent models and eliminating ethical concerns (Ribatti et al. 2010). The CAM has therefore become an indispensable model for the study of vessel development and dynamics in particular, testing anti-angiogenic therapies (Ribatti et al. 1996, 2000). The versatility of the CAM as an experimental model has been extensively reviewed (Ribatti et al. 2016; DeBord et al. 2018; Chu et al. 2021). One of the major benefits to this model is that it can be maintained ex ovo permiting the real time observation of vascular changes in response to various targeted treatments including radiation therapy (review by Mapanao et al. 2021). This made it an ideal system for visualizing MRT-induced changes in vascular permeability.”
Round 2
Reviewer 1 Report
Thanks to the authors for this revised version that integrate my concerns